# Molecular robotic agents that survey molecular landscapes for information retrieval

Sungwook Woo [1,2,3] ✉, Sinem K. Saka [1,2,4], Feng Xuan[1,2,5] & Peng Yin [1,2] ✉

DNA-based artificial motors have allowed the recapitulation of biological functions and the creation of new features. Here, we present a molecular robotic system that surveys molecular environments and reports spatial information in an autonomous and repeated manner. A group of molecular agents, termed 'crawlers', roam around and copy information from DNA-labeled targets, generating records that reflect their trajectories. Based on a mechanism that allows random crawling, we show that our system is capable of counting the number of subunits in example molecular complexes. Our system can also detect multivalent proximities by generating concatenated records from multiple local interactions. We demonstrate this capability by distinguishing colocalization patterns of three proteins inside fixed cells under different conditions. These mechanisms for examining molecular landscapes may serve as a basis towards creating large-scale detailed molecular interaction maps inside the cell with nanoscale resolution.

Building synthetic molecular motors can provide useful insights into the principles that underlie the operations of the natural counterparts and could also provide new functionalities to potential artificial life forms. DNA is an excellent material for designing molecular motors because of its programmability and compatibility with biological systems. DNA-based motors have evolved from simple manually-operated walkers[1,2] to autonomous walkers[3-6], to walkers with coordinated legs[7,8], and to robots that perform tasks such as cargo transport[9], assembly[10], and sorting[11]. DNA-based motors were also used to enable new functions, such as stepwise chemical synthesis[12], signal amplification[13], and controlled drug release[14].

Human knowledge has advanced through explorations of unknown realms across scales, which often involved sending 'agents' to territories that are hard for humans to directly reach. For example, Mars rovers were dispatched to the distant planet to explore its environments. In the Internet space, there are web crawlers to visit and index web pages that are overwhelmingly too many for humans to manually handle. A molecular robot that similarly roams around and collects information from molecular environments may enable new capabilities in examining molecular systems. For example, the ability to survey spatial arrangements at the molecular scale could help refine studies of spatial transcriptomes[15] and proteomes[16]. In combination with the analysis of temporal cues, such capability could also help expand our understanding of spatiotemporal dynamics of intracellular components.

As a natural information-encoding molecule, DNA has allowed molecular engineering for nanoscale assemblies[17-24] and dynamic functions[25-30]. With polymerases and other enzymes added to the toolset, DNA-based systems have been further engineered to program computation[31], reaction networks[32,33], directional walking mechanisms[3-6], and rotational motors[34], and to concatenate arbitrary DNA sequences for multiple functions[35]. DNA-bound microparticles along with enzymes were used to create microscopic agents for reaction networks[36], but the system did not support molecular resolution nor information transfer to a researcher other than through fluorescence signals. Additionally, DNA sequencing-based

[1]Wyss Institute for Biologically Inspired Engineering, Harvard University, Boston, MA 02115, USA. [2]Department of Systems Biology, Harvard Medical School, Boston, MA 02115, USA. [3]Department of Materials Science and Engineering, Pohang University of Science and Technology, Pohang 37673, Korea. [4]Present address: European Molecular Biology Laboratory (EMBL), Heidelberg, Genome Biology Unit, 69117 Heidelberg, Germany. [5]Present address: Spear Bio Inc., Woburn, MA 01801, USA. ✉e-mail: swoo@postech.edu; py@hms.harvard.edu

spatial reconstruction methods using amplicon diffusion[37] or amplicon colonies[38] have been introduced. These methods elegantly showed the potential of using DNA-based systems to encode spatial information, but they require either thermocycling or manual handling for operation and exhibit limited resolutions that remain at the cellular or micrometer scale.

DNA has also been used to study biomolecular interactions based on proximity detection: examples include proximity ligation[39] and extension[40] assays, along with more recent systems that use sophisticated probe designs and readouts such as proximity imaging methods[41,42], proximity-dependent hybridization chain reactions[43], a stamping-like method based on pre-designed templates[44], and proximity nicking[45]. Most of these methods are 'destructive', however, in the sense that each probe can only be used once, hence potentially leaving 'dead spots' and resulting in incomplete analyses (e.g., in an arrangement A-B-C, when A-B is read, C can remain as a dead spot; the proximity nicking[45] method and another recent ligation-based method[46] cleverly circumvent this limitation by activating all nearby spots surrounding a central probe or by 'rescuing' unligated spots, respectively, but they are still destructive methods in that probes once used cannot be reused for subsequent assays with different partners when needed). Researchers developed non-destructive methods[47,48], allowing repeated recording of proximity relationships between different partners, hence enabling near-complete elucidation of spatial organizations after subsequent reconstruction processes. However, the measurements in these methods are typically limited to only 'pairwise' interactions, making direct detection and analysis of interactions involving multiple biomolecules challenging, while most biological processes require interplay between many molecular components[49].

Here we report a molecular robotic system that can record spatial information from molecular landscapes, based on a mechanism of copying information from targets in a sequential fashion. We describe our system as a group of 'agents', as our molecular robots collect information and report it back to us, without us having to directly examine the targets by either taking them out of their natural environment or immobilizing them to special substrates for manipulation (e.g., optical tweezers). The use of agents allows the target molecules to be kept intact and the information to be sampled repeatedly. The agents operate on multiple targets at the same time, each autonomously, reporting back the collection of information. Each agent uses a unique molecular mechanism that allows direct examination of DNA-labeled molecular targets in situ across multiple targets in proximity, in an autonomous, non-destructive, and repeated manner. Our scheme generates a molecular entity that dynamically trails and grows along DNA probes, which we termed a 'molecular crawler' (Fig. 1a). When finished with crawling, the molecule serves as a 'record' that reflects the trajectory and contains information copied from the target-bound probes. We demonstrate two key capabilities for quantitative analyses with crawlers: counting the number of subunits in molecular complexes and detecting multivalent proximity interactions.

## Results

### Design of the crawler system

Each of the probe molecules, along which a crawler follows and grows, is composed of two domains in our basic design: a primer-binding (PB) domain and a copy-and-release (CR) domain (Fig. 1b). The PB domain takes in a primer which then gets extended by a polymerase into the CR domain (Fig. 1c, top row). The CR domain is a double-stranded motif that allows first copying of an arbitrary sequence sequestered within the domain onto a bound primer and then a spontaneous release of the copied segment under an isothermal condition[35,47]. The bottom of the CR domain is met with a 'stopper' point beyond which the polymerase cannot proceed (implemented by a noncanonical base pair, e.g., iso-dC/iso-dG, or a chemical linker). The CR domain is further divided into two subdomains: a barcode domain and a primer-encoding (PE) domain. The barcode domain contains a sequence that identifies the specific probe, which can also serve as a spacer for a tunable reach, depending on the purpose. The PE domain holds a sequence that encodes a primer for a subsequent step. When another probe with a complementary PB domain is nearby, the newly

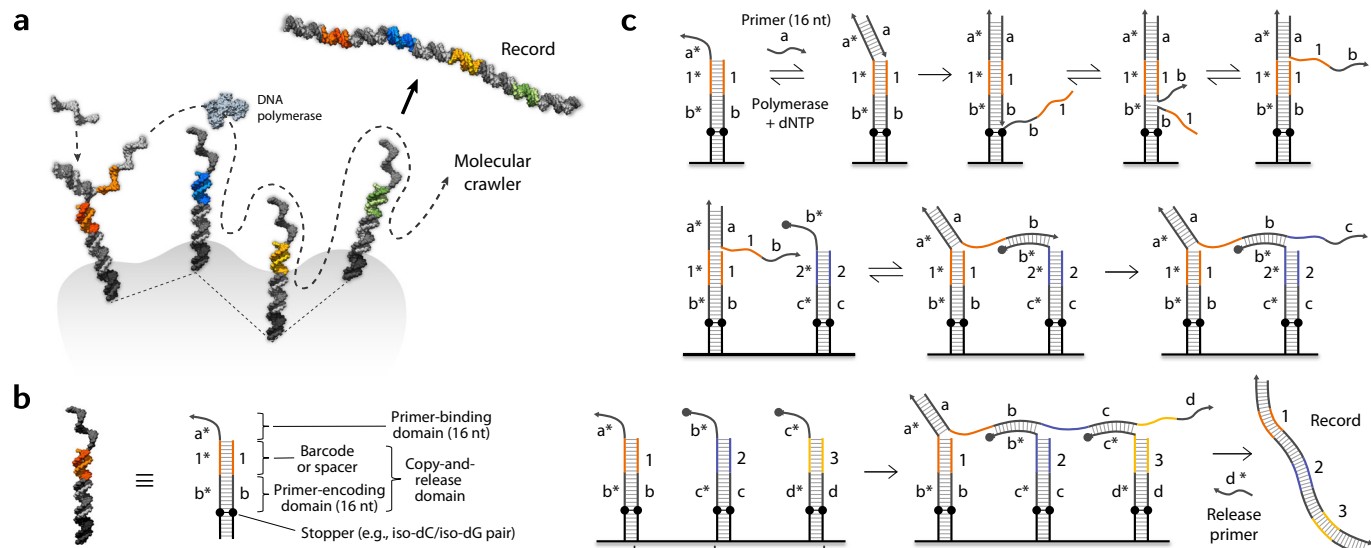

**Fig. 1 | Design and mechanism of the molecular crawler system. a** Schematic showing the crawler concept. A crawler roams around a molecular landscape and generates a record that reflects the trajectory. **b** Anatomy of a probe. See text for details. **c** Basic mechanism of operation. The top row depicts the unit operation in a single probe. A primer (strand 'a') binds the primer-binding domain (a*) and gets elongated by a polymerase along the template. The newly synthesized part competes with the existing strand and can be displaced, exposing a new primer (domain 'b'); the a–a* pair (16 bp) is stable ($T_m \approx 60\,°C$) at the operating temperature (room temp.) and remains bound. The new primer can initiate a next reaction, as shown in the middle row, with another probe nearby (typically within tens of nanometers; tunable). When three probes are in proximity as in the bottom row, a series of reactions yields an extended crawler spanning across the three probes. Upon binding and extension of a release primer, a record can be released into the solution, which also returns the probes to their original state.

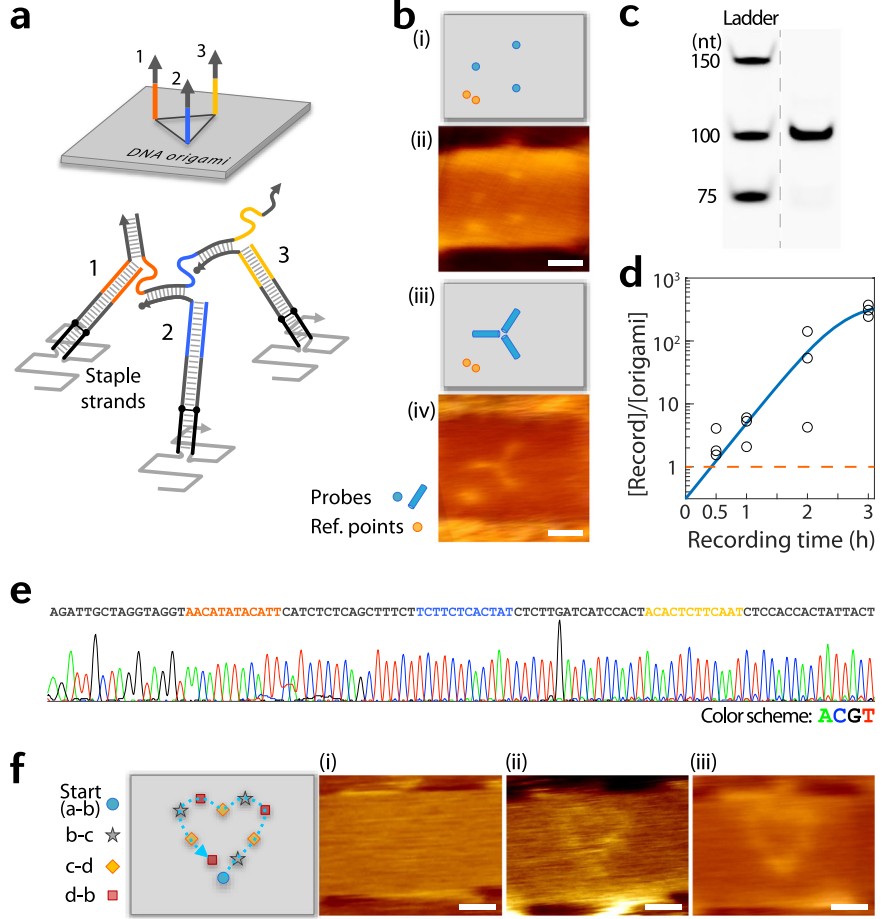

**Fig. 2 | Basic demonstration. a** Schematic of a three-point track created on DNA origami for basic tests. **b** Models and averaged (across different molecules) AFM images of before (i, ii) and after (iii, iv) a crawling reaction. Before reaction, probes appear as dots because the sweeping action of an AFM tip pushes around the tethered probes and can only capture faint images of the anchor points. After ~1 h reaction, but without the release primer added, the crawler now connects and holds the three probes together limiting their movements, and thus appear accordingly in the AFM image. Ref., reference. **c** Gel confirms the correct length of the full record. Lanes were rearranged to allow easy comparison; see Supplementary Fig. 4 for full gel. nt, nucleotides. **d** Record generation quickly exceeds the added origami amount (marked by the dashed orange line) through the non-destructive and repeated recording mechanism. **e** The full record sequence was confirmed by Sanger sequencing. **f** Extended track with ten probes, with repeated arrangements of 'b–c', 'c–d', and 'd–b' probes after the Start (a–b) site, demonstrating the scalability of the crawling reaction. Representative AFM images before (i) and after (ii) a reaction (with no release primer added), along with an averaged (across different molecules) image (iii). Scale bars, 25 nm. Source data are provided as a Source Data file.

generated primer can bind the next probe and initiate 'crawling' (Fig. 1c, middle row).

At the end of a recording round, a 'record' molecule gets autonomously released into the solution, and this allows one of the key properties of the crawler system that the recording process is non-destructive to the probes, leaving the state of each probe unchanged after each round, hence allowing catalytic, repeated recording. This feature is enabled by an additional DNA species in the solution, which we call a "release primer" (strand 'd*' in Fig. 1c, bottom row). When its complement is exposed, the release primer can bind it and get extended by a polymerase all the way to the end (a*), spontaneously releasing the double-stranded record into the solution and returning the probes to the original state. This process also naturally allows retrieval of the records by simply collecting the supernatant solution.

To demonstrate the basic mechanism of the molecular crawlers, we built a three-point track in triangular alignment on a DNA origami[19] platform (see Methods for experimental procedures, Supplementary Figs. 1 and 2 for the sequence diagram and probe position maps, respectively, and Supplementary Data 1–3 for sequence lists). Figure 2a depicts a schematic of the design (top) and a rough diagram of a crawler after crawling over the three probes (bottom). DNA origami

were deposited on a mica substrate to prevent damage by the polymerase[47] and to allow atomic force microscope (AFM) characterizations. Figure 2b shows models and averaged (across different molecules) AFM images of the origami before and ~1 h after crawlers were allowed to trail the three probes (see Supplementary Fig. 3 for AFM images prior to averaging). The fully grown crawler record reaches a length of 100 nucleotides. When the records were retrieved, amplified by PCR and run on a denaturing gel, the final records appear at the expected length range (Fig. 2c; see Supplementary Fig. 4 for full-height gel images). A qPCR analysis shows that record generation quickly (within ~30 min) surpassed the number of origami templates added and reached a ~100-fold excess in 3 h (Fig. 2d) under a certain recording condition, which supports the non-destructive and repeated recording mechanism. The sequence of the records was also confirmed using Sanger sequencing (Fig. 2e).

We further demonstrated the scalability of the crawling action beyond three probes, by using a track with ten prescribed sites. To prescribe each step, the primers for each probe would normally have to be orthogonal to each other. However, since the track sites are spatially restricted, not all primers have to be fully orthogonal to one another; i.e., for sites that are far enough in space to interfere, the same primers

can be reused. Hence, repeated alignments of a set of distinct probes along a desired path can allow multiple steps in a prescribed manner. Figure 2f shows such an example where a set of three kinds of probes (each marked 'b–c', 'c–d', and 'd–b') are placed along the boundary of a shape in order and in a repeated fashion, with a crawler successfully grown along the track. See Supplementary Fig. 5 for an additional set of data that supports scalability of the crawling reaction.

## Random crawling and counting

To allow general applications, it is desired for the crawler system to have the ability to randomly move around and examine unspecified targets, rather than in a fully prescribed fashion as shown above. We achieved a random crawling behavior by introducing a mechanism based on a single type of 'universal' probe, which allows the initiation, crawling and release of a crawler at 'any' site. Such a universal probe can be designed by incorporating a 'tandem' PB domain and 'repeating' primers. A tandem PB domain has two PB sites, e.g., 'a*' and 'b*', concatenated together (Fig. 3a), such that it can take in either primer 'a' or 'b'. Repeating primers denote the arrangement of primers and PB sites such that the incoming primer and the outgoing primer on a single probe are the same (the domains marked 'b'/'b*' in the example shown in Fig. 3a). When an initiating primer 'a' binds the 'a*' section of the tandem PB domain and gets extended through the probe, a new domain 'b' is generated at the bottom. The new 'b' can then act as a primer for a crawling reaction to a nearby probe by binding the 'b*' segment of the tandem PB domain of the next probe. Since each universal probe has the 'a*' domain, the crawling reaction can initiate at 'any' site. Since 'b' is the connecting primer that allows crawling between probes, the crawler can proceed to 'any' site in proximity. Using strand 'b*' as a release primer, the crawler can be released at 'any' site as well. The combination of these rules allows a truly random crawling mechanism. The molecular detail is shown in Fig. 3b. (Note we cannot use 'b' as an initiating primer as 'b' and the release primer 'b*' will cancel each other out in solution).

This random crawling principle naturally allows a mechanism for 'counting'—counting the number of subunits within a molecular complex, because the longest possible record generated from a molecular complex reflects the maximum number of steps that can be taken within the complex and in turn the number of subunits. We first demonstrate this capability by using a model system based on streptavidin. Streptavidin is a well-known tetrameric protein, where each monomer contains one binding site for biotin. We incorporate a universal probe that contains a tandem PB domain (b*-a*) and encodes 'b' as a new primer (collectively, denoted 'a-b-b') at each biotin pocket through a biotinylated DNA strand as shown in Fig. 3a. An initiating primer 'a' can start a reaction at any of the four sites, then crawl to any of the nearby sites if available or be released by a release primer ('b*') at any site. This allows four kinds of records to be generated, each with a distinct length. The shortest one is a 'partial record' from only one of the probes, released by a release primer before the crawler was able to crawl to a nearby probe. The longest one is a 'full record' generated by a crawler that visited all four probes on a single tetramer. There are two intermediate-length records that are created when a crawler visited two or three subunits on a tetramer before getting released. The four kinds of records with distinct lengths show up as distinct bands on a gel as shown in Fig. 3c (left lane, marked '+SA'). In the absence of streptavidin, the crawler can count only up to '1', as the probes are freely floating in solution without forming a complex, as shown in the right lane (marked '−SA'). The shortest record that corresponds to 'count one' is made from a single probe whether or not it is bound to a complex, as expected.

We further demonstrate the counting capability in a more programmed fashion, by using DNA origami to create artificial molecular complexes with tunable size. Since recording on surface-bound DNA origami requires PCR amplification due to the typically low

concentration, we assigned special probes on the start site ('a-b') and the finish site ('b-d') such that we can amplify the final products with primers 'a' and 'd*'. Between the start and finish sites, we set four variable positions. For these positions, we used probes with repeating primers ('b-b') and varied the inclusion of these probes (e.g., from none to all four) to change the size of the complex from two to six in total, including the start and finish sites (Fig. 3d). Figure 3e is the schematic diagram and Fig. 3f shows the gel data revealing distinct bands for the counts for each complex. This example also highlights the power of crawlers as molecular 'agents', as the crawlers take the measurements from a low concentration sample then amplify the results for easy readout.

One may note that, in random crawling, a crawler would take a different path each time, as each step involving repeating primers can occur in any direction. Thus, records with the same length produced after the same number of steps can contain information for different trajectories. To test if we can 'decode' the different paths, we designed a simple square-shaped track containing two probes with repeating primers, each labeled with different barcodes (Fig. 3g), and analyzed the records with high-throughput sequencing. Gel data shows that crawlers produce records of three different lengths, as expected from a complex of size 4 with start and finish sites, but both length-3 (len3) and length-4 (len4) records can originate from two different paths, respectively, as depicted in Fig. 3h. Sequencing analysis confirmed that it is indeed the case, and revealed the distribution of different paths taken (Fig. 3i, j). While the ratios between sequencing read counts do not directly represent the actual ratios of record populations especially between different length products, due to potential biases in PCR and sequencing processes, the general trends within len3 and len4 products display the 'preference' of the crawlers for a shorter path over a longer (diagonal) one, likely reflecting kinetic advantage.

## Multivalent interaction detection

The property of molecular crawlers that they create concatenated records from proximal interactions regardless of the number of constituent components makes them a powerful tool for detection of multivalent proximal interactions. Multivalent interactions can be detected directly in situ by observing the generation of the corresponding records, or the records can be collected and further subjected to diverse post-processes such as gel analysis or sequencing. Previous attempts to detect colocalization of multiple target species include approaches based on ligation[50], templating[44], and nicking[45]. In addition to being destructive to probes as described earlier, these methods have either limited throughput due to requiring imaging or limited generalizability due to relying on pre-designed ligation connectors or structural templates; in particular the ligation mechanism required multiple probe species to bind together in a fixed configuration and in a stoichiometry-sensitive manner, which limits scalability. Our DNA probes need not bind together simultaneously in a predefined way to detect multiple interactions, as each step only requires two proximal partners to interact at a time. As each step is local and independent of previous steps regardless of the total number of components, our mechanism is highly scalable in terms of the number of detectable interactions; we showed a ten-probe example above and a crawler can in principle keep growing as long as there are interaction partners in proximity and resources (e.g., nucleotides) are supplied (see Supplementary Fig. 5). In addition, since the recording mechanism is non-destructive and catalytic, multiple rounds of recording can occur on the same target, hence amplifying signals that might otherwise remain negligible.

We applied the mechanism to examining the colocalization of three protein species at the microtubule growing end inside fixed cells, preserving the in situ physiological context. We detected the trivalent colocalization interaction of alpha-tubulin, beta-tubulin, and end-binding protein 1 (EB1). While alpha and beta tubulins form

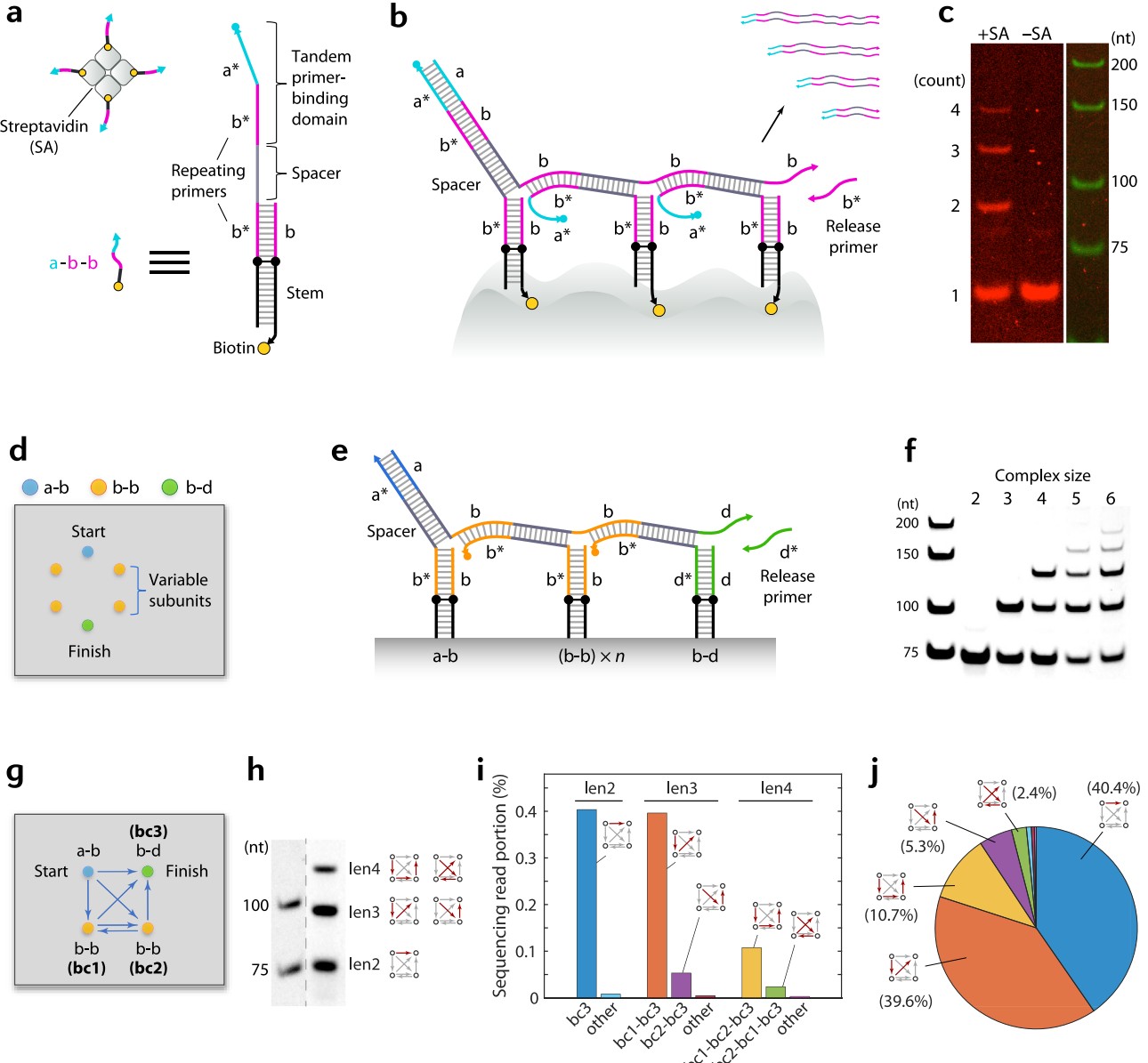

**Fig. 3 | Random crawling mechanism and counting. a** Schematic and detailed diagram of the universal probe, designed to allow the initiation, crawling and release of a crawler at any site, thereby enabling random crawling. **b** Strand diagram showing a crawler over three universal probes as an example, along with a schematic depicting four kinds of possible records from streptavidin. **c** Gel image showing the results of 'counting' the number of subunits in streptavidin (+SA) and in its absence (−SA). Ladder lane is from a different gel with the same '+SA' sample run in parallel; see Supplementary Fig. 4 for sample lane next to ladder. nt, nucleotides. **d** Schematic showing artificial molecular complexes with tunable size designed on DNA origami. **e** Strand diagram showing a crawler over three probes; the middle probe (b–b) is a variable unit. Actual probes used had an alternative architecture (Supplementary Fig. 6), but diagram here was drawn with the basic design for clarity. **f** Gel image showing the counting results for the artificial complexes. Note the number of bands is the complex size minus one in this design. **g** Schematic of a square-shaped track for path 'deconvolution' tests. The middle probes with repeating primers (b–b) had barcodes embedded to be used to distinguish different paths by high-throughput sequencing. **h** Gel data showing possible records with three different lengths. Records of length-3 (denoted len3) and of 4 (len4) can be produced from two different paths, respectively. Lanes were rearranged to allow easy comparison; see Supplementary Fig. 4 for full gel lanes or Source Data for full gel. The sample lane is from a Cy5 channel scan. **i** Sequencing analysis reveals the relative populations of different paths, unveiling the 'preference' of the crawlers for shorter paths. **j** Sequencing read portions replotted across all record lengths for easy comparison. Source data are provided as a Source Data file.

heterodimers that constitute microtubules and thus are found ubiquitously along microtubules and inside the cell, EB1 only interacts with the growing end of a microtubule, supporting its stabilization and recruitment of other proteins[51]. If a microtubule experiences catastrophic disassembly or otherwise goes out of the growing phase, EB1 dissociates from the microtubule. We targeted the three proteins using a mixture of three respective antibodies which are labeled either directly or indirectly (through secondary antibodies) with orthogonal

DNA anchors, to which crawler probes specific to respective proteins were allowed to bind. The probes were designed such that recording starts when the initiating primer 'a' binds the 'a*' domain of the 'a–b' probe on alpha-tubulin, then proceeds to the 'b–c' probe on beta-tubulin, and then on to the 'c–d' probe on EB1 (Fig. 4a). In the presence of a release primer 'd*', the trivalent records—and only they—can be collected and analyzed. In separate recording rounds, if different primer sets are used, the presence of each protein monomer can be

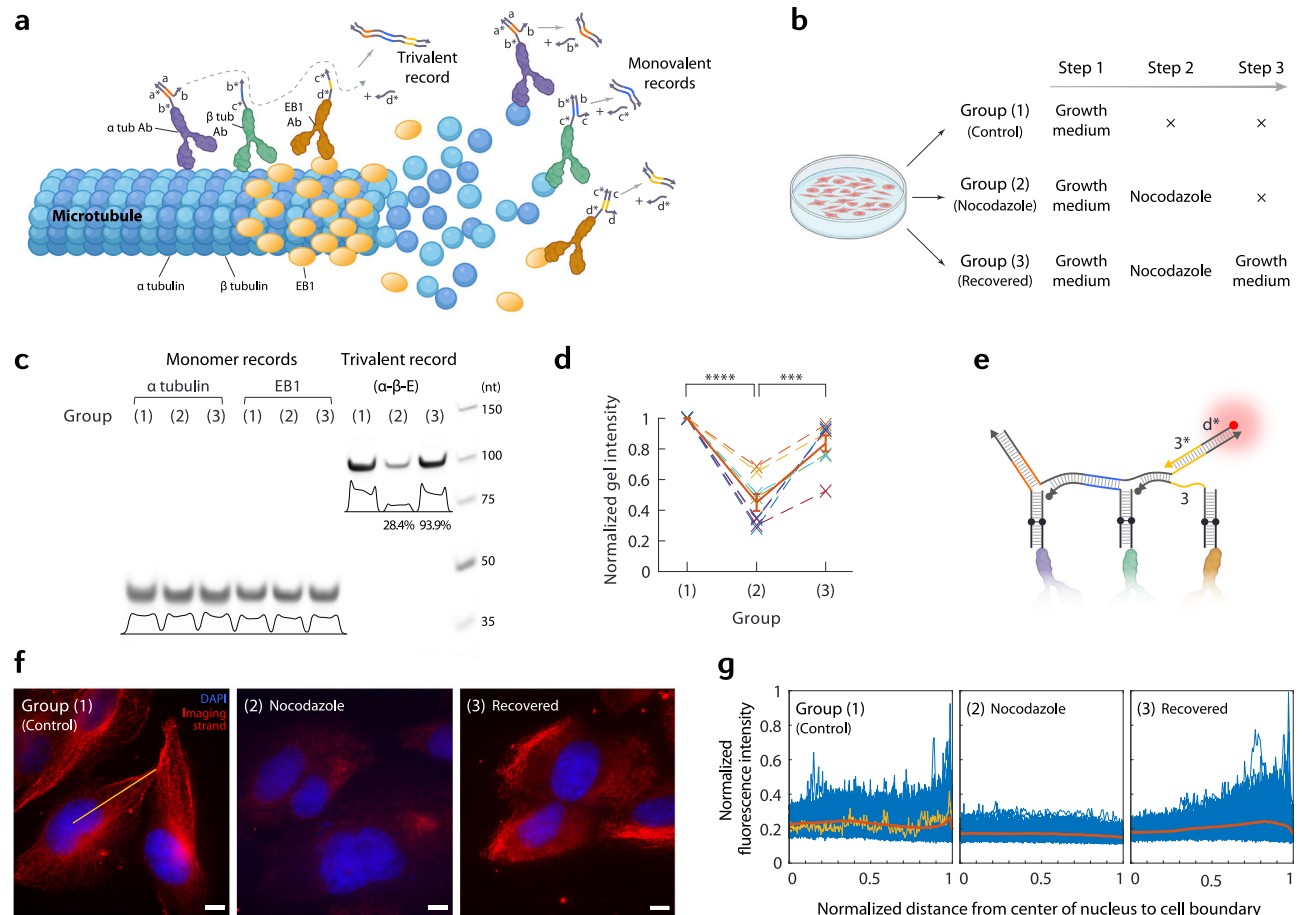

**Fig. 4 | Multivalent interaction detection. a** Schematic showing the setup and mechanisms of trivalent and monovalent records. Note for β tubulin and EB1, labeling used secondary antibodies but was simplified here for clarity. **b** Three groups of cell populations and their respective treatments. **c** Gel image showing two kinds of monovalent records and the trivalent record for the three groups. While there is no noticeable difference in the level of monovalent records between groups, the level of trivalent records drops for group (2) and recovers for group (3). nt, nucleotides. **d** Plot of normalized gel intensities of trivalent records for multiple experiments ($n = 8$; dotted lines) along with the average (solid line). Error bars are the standard errors of the means. ****$P \le 0.0001$, ***$P \le 0.001$. **e** Schematic showing

a crawler over three probes and a fluorescent strand used for imaging confirmation. Note we used strand 'd*−3*' for increased stability. **f** Representative fluorescence images showing the characteristic differences between the three groups. DAPI stains nuclei. The straight yellow line is an example trace for intensity profile measurement; see **g**. Scale bars, 10 μm. **g** Fluorescence intensity profiles (example trace and its profile shown in yellow in **f** and **g**, respectively, for Group (1)) normalized both in the distance and in intensity (average profiles in orange) show peaks near the periphery of cells in Groups (1) and (3) only. Source data are provided as a Source Data file.

independently detected: e.g., 'a' and 'b*' can generate a short monovalent record from the probe bound to alpha-tubulin, 'b' and 'c*' from beta-tubulin, and 'c' and 'd*' from EB1, respectively. Depending on the microtubule growth state of the cell, the relative levels of the trivalent and monovalent records would show up differently; e.g., in an actively growing state, the level of trivalent records is expected to increase.

To illustrate the ability of the crawler system to distinguish different cellular states by detecting protein colocalizations, we designed a perturbation experiment with three conditions (Fig. 4b). The first group, (1), was treated with a normal growth medium, where cells would exhibit typical microtubule polymerization dynamics. We expect to see both trivalent and monovalent records to be significant in this state. The second group, (2), was treated the same way as group (1), then subsequently treated with nocodazole, a drug that promotes microtubule disassembly[52], for 1 h at 37 °C. For this group, due to the effect of the drug, the level of trivalent records is expected to drop; however, since the monomers, both the tubulins and EB1, are still present within the cells, the level of monovalent records is expected to remain unchanged. The third group, (3), was treated the same way as group (2), then additionally was treated again with a growth medium for 1 h such that the cells recover from the drug and form microtubules

again. In this state, the level of trivalent records is expected to return to a significant level, while the monovalent records would still be comparable. After recording for trivalent and monovalent records for each group, and PCR amplification, the results shown in gel are highly consistent with these expectations (Fig. 4c, d): while the level of monovalent records for alpha-tubulin and EB1 remained nearly the same throughout the steps, the level of trivalent records dropped significantly for group (2) and recovered to a level comparable to group (1) after the regrowth treatment for group (3) (28.4% and 93.9%, respectively, of group (1), comparing gel band intensities, in the particular test shown in Fig. 4c; or ~4.7% and ~68.9%, respectively, in pre-amplification record quantity, estimated based on gel-based calibration tests—see Supplementary Fig. 7).

While the gel results successfully proved the ability of the crawler system to detect multivalent protein interactions and to distinguish different growth states of cells, we further sought to confirm whether the records were indeed created at the right places, by directly observing the crawler-generated records in situ. As only the trivalent records will expose the final primer 'd', by adding a fluorescently labeled strand complementary to 'd' (Fig. 4e), we can spatially localize the trivalent records that were just created in situ. The fluorescence

microscopy results are as expected (Fig. 4f). For group (1) cells, which are in an actively growing state, we observed strong fluorescence signals appearing near and along the periphery of the cells, whereas for group (2) cells, where the drug disassembled microtubules, hence there is no growing microtubule ends, fluorescence signals were barely detectable or remained at the background level. After the recovery treatment, group (3) cells restored the growth state and exhibited active growing ends again near and along the periphery of the cells. We took intensity profiles of the fluorescence signals along the axis from the center of the cell nucleus to the boundary (Fig. 4g). While, in group (1), a signal peak is observed near the cellular periphery, in group (2), the signal is flat and lower in intensity (~74% of the basal level of group (1)). In group (3), we clearly see the signal being restored towards the cell periphery.

## Discussion

We have demonstrated a system of molecular robotic 'agents' for quantitative examination of molecular landscapes. The use of agents allowed non-destructive and repeated sampling of information from target molecules, enabling amplification of weak signals from low concentration samples while the molecules are kept intact. While the task of each agent was simple and minimal, their combined products allowed us to reach answers that one agent cannot directly provide, such as the number of subunits in molecular complexes and the status of multivalent protein colocalization. With more development, some form of 'swarm' behaviors could also be designed, by encoding 'interactions' between the agents, e.g., through a record generated by one agent becoming an instruction to be read and followed by another agent.

Quantitative understanding has been crucial in biological studies; e.g., the three-state model[53] for the operation of ATP synthase, which contributed significantly to our understanding of the machinery, had been proposed primarily based on quantitative knowledge of reaction products, before the crystal structure was solved[54]. Membrane proteins typically form clusters of certain sizes when initiating signal cascades, thus it is often important to find the critical size of clusters to understand signal pathways and mechanisms[55]. We expect that our crawler system would find a wide range of uses in quantitative biological studies, especially where protein extraction or imaging-based methods are not easily applicable, e.g., for high-throughput analysis of low-copy-number proteins that perform critical functions in biology[56].

The crawler system was anchored to target molecules through DNA tagging, which was achieved, in this study, by direct DNA hybridization, streptavidin-biotin interaction, or antibody conjugation. Additional labeling modes, such as genetic tagging followed by chemical coupling or the use of aptamers or nanobodies, should in principle be compatible and would allow broader applications. DNA hybridization could also be exploited to directly target different parts of chromosomal DNA, potentially allowing studies of protein-DNA interactions and of 3D chromosome organizations. In the long run, with unique labeling and massively-parallel sequencing readouts, the crawler system could provide a route towards recording interactions of numerous individual components, e.g., inside a fixed cell, and thus allowing an extensive map of molecular interactions to be created with nanoscale resolution.

## Methods

### DNA oligonucleotides

DNA oligos were purchased from Integrated DNA Technologies (IDT). Most origami staples were ordered unpurified (standard desalting; STD), except some staples with probe extensions, which were purchased purified either by PAGE or HPLC (details available in the sequence list in Supplementary Data 2). Similarly, most probes and fluorescently labeled strands were ordered purified either by PAGE or HPLC (Supplementary Data 3). All oligos were purchased

pre-suspended at 100 μM in TE buffer (10 mM Tris, 0.1 mM EDTA) and were stored either at 4 °C or −20 °C. DNA origami scaffold strand (single-stranded M13mp18 DNA) was purchased from Bayou Biolabs (Cat. No. P-107).

### DNA origami preparation

The design of the 2D rectangular origami was adapted from a previous study[47]. Scaffold and regular staples were mixed together at target concentrations of 5 nM and 40 nM, respectively, in TAE (40 mM Tris-acetate, 1 mM EDTA) buffer supplemented with 12.5 mM magnesium acetate (referred to as "TAE/Mg"). Staples with probe extensions (see Supplementary Fig. 6a) were added at 1.5× (5' side) and at 1.8× (3' side) of regular staples. Staples containing anchors for probes (see Supplementary Fig. 6b) were added at 1.33–2× of regular staples, the bottom probes ('adpt') at 1.5-2× of the anchor staples, and the top probes at 1.2–2× of the bottom probes. For origami folding, the mixture was kept at 90 °C for 5 min and annealed from 90 to 60 °C at a rate of −1 °/min, then from 60 to 45 °C at a rate of −1/6 °/min (0.1 ° drop every 36 sec), and then from 45 to 25 °C at a rate of −1 °/min. Folded origami solutions were used without further purification unless otherwise noted.

### Streptavidin sample preparation

First, probe duplexes were created by mixing the biotinylated anchor strand and the probe strand at 1 μM and 1.2 μM, respectively, in 1× TAE/ Mg, and heating the mixture to 90 °C for 5 min, then cooling it from 90 to 25 °C at a rate of −1 °/min. The probe duplexes were then incubated with streptavidin (Life Technologies, Cat. No. 434302) at 80 nM and 20 nM, respectively, in buffer A (10 mM Tris pH 8, 100 mM NaCl), at 37 °C for 1 h. A control sample without streptavidin was prepared alongside by adding buffer A in place of streptavidin, with all other conditions kept the same.

### Cell sample preparation

For α-β-EB1 colocalization tests, BS-C-1 cells (ATCC CCL-26, African green monkey) were grown in DMEM, high glucose, GlutaMAX Supplement (Gibco, 10566016) supplemented with 10% (v/v) serum (Gibco, 10437), 50 U/ml penicillin and 50 μg/ml streptomycin (Gibco, 15070) at 37 °C in the presence of 5% $CO_2$. The cells were plated on eight-well glass-bottom μ-slides (ibidi, 80827) and further grown for 5 days to 50–60% confluency. Three groups of cells were treated in parallel, with each step applied in a 'staggered' manner such that the treatments for the three groups end at the same time. First, Group (3) samples were treated with nocodazole by replacing the growth medium with 250 μl of pre-warmed media containing 2.5 μg/ml (~8.3 μM) of nocodazole (diluted from a 1000× stock in DMSO; Sigma, M1404) and incubating the samples for 1 h at 37 °C. Then the nocodazole solution was replaced with 250 μl of a fresh warm medium, and the cells in Group (3) samples were let recover for 1 h at 37 °C. At the same time, nocodazole treatment for Group (2) samples was initiated. Since nocodazole stock was in DMSO, to compensate for any potential carrier effect, Group (1) samples were also treated with cell medium containing 0.1% DMSO in parallel. After treatments, cells were gently washed with PBS (Invitrogen AM9625, 1× solution diluted from 10× stock) twice and were fixed with 4% paraformaldehyde (PFA, diluted from 16% solution, Thermo Scientific #28908) for 45 min, quenched with 100 mM $NH_4Cl$ in PBS for 15 min and washed with PBS for 5 min. Cells were then permeabilized and blocked in 2% nuclease-free BSA (AmericanBIO, CAS 9048-46-8) with 0.1% Triton (Sigma-Aldrich T8787) in PBS for 30 min.

Samples were then incubated with primary antibodies overnight at 4 °C using the following antibodies and dilutions: b0-DNA-conjugated rat alpha-tubulin antibody (Invitrogen, Cat. No. MA1-80017, Clone YL1/2) at 1:250, mouse beta-tubulin antibody (Developmental Studies Hybridoma Bank, Cat. No. E7, Clone E7) at 1:200, and rabbit EB1

antibody (Sigma, Cat. No. E3406, polyclonal) at 1:750, in incubation buffer consisting of 0.1% Triton X-100, 2% nuclease-free BSA, 0.2 mg/ml sheared salmon sperm DNA (Thermo Fisher, AM9680), 0.02% sodium azide and 4 mM EDTA (Ambion, AM9260G) in PBS. Then the samples were washed with PBS containing 0.1% Triton X-100 and 2% BSA for 10 min three times. Finally, samples were incubated for 1 h at room temperature (RT) with the following DNA-conjugated secondary antibodies at 1:200 dilution in the same antibody incubation buffer used above: b2-anti-mouse (based on Jackson ImmunoResearch Cat. No. 715-005-151, polyclonal) and b3-anti-rabbit (based on Jackson ImmunoResearch Cat. No. 711-005-152, polyclonal). After the incubation, samples were washed with 2% BSA + 0.1% Triton-PBS for 10 min, followed by washing with PBS for 10 min twice. Then the samples were post-fixed using 5 mM BS(PEG)5 (Thermo Fisher, 21581) in PBS for 30 min, followed by quenching in 100 mM NH$_4$Cl in PBS for 5 min, and kept at 4 °C in PBS supplemented with 500 mM NaCl and 0.02% sodium azide until further use.

All antibodies used are from commercial sources, with validation data including immunofluorescence and immunoblot results and relevant references (at least 20 for each as of March 2023) available from their respective web pages. Following is a list of one reference for each antibody as an example: (1) rat alpha-tubulin antibody—doi: 10.1038/ncomms13874; (2) mouse beta-tubulin antibody—doi: 10.1101/pdb.prot105635; (3) rabbit EB1 antibody—doi: 10.1038/ncomms11665; (4) anti-mouse secondary antibody—doi: 10.1038/s41467-022–32626-0; (5) anti-rabbit secondary antibody—doi: 10.1038/s41467-023–38943-2.

**Antibody-DNA conjugation**

Secondary antibody to DNA conjugations were performed as described previously[57]. In brief, 25 μl of 1 mM 5′-thiol modified DNA oligonucleotides was activated by treating with 100 mM dithiothreitol (DTT) (Thermo Fisher, 20291) for 2 h at RT in darkness, then purified using NAP5 columns (GE Healthcare Life Sciences, 17-0853-02) to remove excess DTT. Antibodies formulated in PBS only were concentrated using 0.5-ml 50-kDa Amicon Ultra Filters (EMD Millipore, UFC510096) to 2 mg/ml and reacted with maleimide–PEG$_2$–succinimidyl ester crosslinkers (Thermo Fisher, 22102) for 1.5 h at 4 °C (mixed at a ratio of 100 μg antibodies to 2.5–3.4 μl of 0.85 mg/ml crosslinker). Reaction mixtures containing antibodies were then buffer-exchanged using 0.5-ml 7-kDa Zeba desalting columns (Thermo Fisher, 89883) to remove excess crosslinkers. Activated DNA oligonucleotides were incubated with antibodies (at an antibody to single-stranded DNA molar ratio of ~1:11) overnight at 4 °C. The final conjugated antibodies were washed using 2-ml 50-kDa Amicon Ultra Filters six times to remove non-reacted DNA oligonucleotides. The DNA oligos were ordered from IDT with 5′ thiol modification (/5ThioMC6-D/) with 2 T residues added as a spacer between the antibody linker sequence and the crosslinking group[57,58]. Detailed sequences are in the sequence list in Supplementary Data 3.

**Crawler recording reactions**

Crawler reactions on DNA origami were carried out as follows. First, flow chambers were created by adhering a freshly cleaved mica (Ted Pella) piece to a channel slide system (ibidi, sticky-Slide VI 0.4, Cat. No. 80608), yielding a 30 μl reaction volume in each chamber. For each reaction, a chamber was first washed three times with 60 μl TAE/Mg by adding the buffer to one reservoir (inlet) and subsequently taking out the same volume from the other side (outlet); after the first washing round, 30 μl of buffer remains in the chamber due to capillary forces and only 30 μl of extra buffer is drawn through the outlet. Then a 30 μl solution containing origami, typically at 50 pM concentration (see below for special cases) was introduced to the chamber by adding the solution to the inlet and drawing the same volume of buffer from the outlet. After a 30 min incubation, unbound origami and extra

staple strands were removed by washing the chamber three times with 60 μl fresh TAE/Mg, followed by washing three times with 60 μl of 1× ThermoPol buffer (New England Biolabs (NEB), Cat. No. B9004S, 10×; 1× contains 20 mM Tris-HCl, 10 mM (NH$_4$)$_2$SO$_4$, 10 mM KCl, 2 mM MgSO$_4$, 0.1% Triton X-100). To the chamber, a 40 μl solution containing Bst polymerase large fragment (NEB, Cat. No. M0275S), dNTP (NEB, Cat. No. N0447S), and relevant primer mixes in 1× ThermoPol reaction buffer was added and incubated for 1 h at RT. The typical concentrations of the components used after optimization were as follows: Bst at between 8 U/ml and 0.8 U/μl, dNTP at 100 μM, primer 'a' at 100 nM, and the release primer at 10 nM. After the recording reaction, the supernatant solution containing product records was collected and was treated with Exonuclease I (NEB, Cat. No. M0293S) by mixing the records and Exonuclease I at a 9:1 volume ratio and incubating the mixture at 37 °C for 20 min, to remove extra recording primers contained in the product solution. Finally, the enzymes in the solution were heat-inactivated by incubating the solution at 80 °C for 20 min.

In some cases, adjustments were made to the reaction mixtures. For the qPCR-based record quantification tests, to help highlight the ratio, an origami solution at 0.5 pM and a recording mix containing Bst, dNTP, primer 'a' and the release primer at 0.8 U/μl, 1 mM, 1 μM and 100 nM, respectively, were used. Samples for different recording durations (0.5, 1, 2 and 3 h) were prepared in separate flow chambers and treated in parallel. The experiments were conducted in triplicate to generate the data points displayed in Fig. 2d. For the counting experiment (as shown in Fig. 3d), a trace amount of an additional primer ('b*') was added at 1 nM to help reset origami where a crawler might be 'stuck' in one of the variable subunits. For the path deconvolution test described in Fig. 3g, a 250 pM origami solution was used to boost the overall record production to reduce the number of PCR cycles to help minimize PCR-related biases in record populations.

For recording under AFM to image the samples before and after recording (as shown in Fig. 2b, f), a 5 μl of origami solution diluted to 2 nM in TAE/Mg was deposited to a small (~1 cm by 1 cm) freshly cleaved mica piece, to which 20 μl of TAE/Mg was added. After first taking 'before' images, extra staples were washed away by taking out as much buffer as possible with a micropipetter (leaving a thin layer of buffer spread on the hydrophilic mica surface) and adding 20 μl of fresh buffer, and repeating this process 6 times in total. Then the buffer was exchanged to 1× ThermoPol buffer by washing with the buffer similarly 6 times. After taking out 20 μl of buffer on top of the mica piece, a 40 μl solution containing recording mix prepared as above (but with only 'a' as a primer) was added and incubated for ~1–2 h at RT.

Crawling reactions for solution samples (e.g., streptavidin) were carried out by mixing the probe duplex (with or without streptavidin), Cy5-labeled forward primer 'a', release primer 'b*', Bsm polymerase large fragment (Thermo Scientific, Cat. No. EP0691), and dNTP at 4 nM, 2.5 nM, 200 nM, 2.5 U/ml, and 2 μM, respectively, in 1× Bsm buffer (typical total volume 10 μl) and incubating the mixture at 37 °C for 2 h. The products were analyzed by gel electrophoresis directly without further processes (e.g., PCR amplification).

For crawling reactions in cell samples, probe duplexes for the 'a–b', 'b–c', and 'c–d' probes, respectively were prepared in advance by mixing the bottom 'linker connector' strand at 6 μM and the top 'probe' strand at 7.2 μM in 1× TAE/Mg, and heating the mixture to 90 °C for 5 min, then cooling it from 90 to 25 °C at a rate of −1 °/min. All three kinds of probe duplexes were then mixed together (to a final concentration of 250 nM each duplex) along with 'linker protector' strands at 250 nM each in 1× TAE/Mg (detailed diagram in Supplementary Fig. 6d), and 120 μl of the solution was incubated with cell samples in each chamber at RT for 1 h on a shaker with a gentle setting, to allow the probes to bind the respective antibodies. After the incubation, the samples were washed with 0.5× TAE/Mg by first removing the solution and adding 250 μl of the buffer, then leaving the samples on a shaker for 5 min, repeated three times. Crawling reactions in cell samples were

performed by first washing the probe-bound cell samples with 1× ThermoPol buffer three times (by removing the existing buffer and adding 250 µl fresh buffer), then by replacing the buffer with a 120 µl reaction mix containing 8 U/ml Bst, 100 µM dNTP and relevant primer mixes at 100 nM (initiating primer) and 10 nM (release primer) in 1× ThermoPol reaction buffer and incubating the sample for 1–2 h at RT on a shaker with a gentle setting, followed by solution retrieval, Exonuclease I treatment and heat inactivation as described above. Cell samples were sometimes 'reset' to release bound probe strands and to be reused by incubating them with 60% formamide in PBS overnight (over 12 h) at 37 °C on a shaker, followed by washing three times with PBS.

### Atomic force microscopy
AFM images were taken in fluid tapping mode with a Multimode AFM (Veeco Metrology Group) using a Nanoscope V controller. Silicon nitride cantilevers with 2 nm radius silicon tips were used (SNL probes from Bruker).

### PCR amplification, gel electrophoresis, Sanger sequencing, and high-throughput sequencing
PCR amplification mixtures were typically prepared by combining 3 µl of a record solution and 9 µl of a PCR master mix that contained Vent (exo-) DNA polymerase (NEB, Cat. No. M0257S), dNTP, and relevant primers. The concentrations of the components in the final mixture were 40 U/ml Vent, 200 µM dNTP, and 200 nM each primer in 1× ThermoPol buffer. PCR protocol was as follows: activation at 95 °C for 2 min, 20 cycles of (denaturing at 95 °C for 15 sec, annealing at 56 °C for 30 sec, and extension at 92 °C for 15 sec), followed by final extension at 72 °C for 2 min and cooling to 4 °C. For qPCR measurements, which were run in a QuantStudio 1 machine (Thermo Fisher), EvaGreen dye (BioFACT, Cat. No. 31000-B500; 20×) was included at 1× concentration in the final PCR mixture, and the PCR cycle number was increased to 45, with a fluorescence measurement step added after the extension step of each cycle. Standard samples for qPCR quantification were prepared by serially diluting the reference record duplex (ordered in full length) in TE buffer to result in tubes with 0.1 fM through 100 pM templates with 10-fold increments. The fit to the data points in Fig. 2d was obtained using the 'logistic' sigmoidal model in MATLAB ($y = 408.5 / (1 + \exp(-2.77(x-2.591)))$, $R^2 = 0.9216$).

For denaturing polyacrylamide gel electrophoresis (PAGE), gels were cast in-house by mixing 8% acrylamide (19:1 acrylamide:bisacrylamide solution; J. T. Baker Cat. No. 4968-00, IBI Scientific Cat. No. 70014, Crystalgen Cat. No. 221-002-19, or HanLab Cat. No. HAB-0105-40), 7 M urea (Sigma-Aldrich, Invitrogen Cat. No. AM9902, Santa Cruz Biotechnology Cat. No. sc-29114, or GeorgiaChem Cas. No. 57-13-6), 0.06% TEMED (Thermo Scientific, Cat. No. 17919 or GeorgiaChem Cas. No. 110-18-9), and 0.06 % (w/v) ammonium persulfate (Sigma-Aldrich or ACROS organics; Cas. No. 7727-54-0) in 1× TAE (prepared with 50×; Growcells Cat. No. MRGF4210 or T&I Cat. No. BTA-9050) buffer and pouring the mixture into gel cassettes (Invitrogen, Cat. No. NC2015) capped with gel combs (Invitrogen, Cat. No. NC3515), then waiting ~40 min at RT. Gels were typically run for ~30 min under the voltage of 200 V at 65 °C in 1× TAE. Gels were then stained by shaking in a 1× SYBR Gold (Invitrogen Cat. No. S11494) solution in 1× TAE for 20 min, then imaged by Typhoon FLA 9000 (GE), FluorChem M (ProteinSimple) or Azure C600 (Azure Biosystems). The ladder used in gels is GeneRuler Ultra Low Range DNA Ladder (Thermo Scientific, Cat. No. SM1213). Uncropped gel images are included in the Source Data file.

For Sanger sequencing, the products from the initial PCR above were first treated with ExoSAP-IT Express (Applied Biosystems, Cat. No. 75001) according to the manufacturer's suggested protocol to remove any remaining primers and dNTP, then diluted 100-fold in nuclease-free water before being subjected to the next rounds of PCR using adapter primers. PCR with adapter primers was performed using 1× Q5

DNA polymerase (NEB, Cat. No. M0492) with all other conditions kept the same as above, followed by an ExoSAP-IT treatment. The products were then diluted again 100-fold in water, and PCR-amplified using a pair of shorter primers and Q5, followed by an ExoSAP-IT or Exonuclease I treatment as above. The last round of PCR was optionally repeated to improve purity. Sanger sequencing was performed by Genewiz, Inc.

For high-throughput sequencing (next-generation sequencing; NGS), recording products were first PCR-amplified as described above, but using relevant adapter primers and Q5 polymerase, and with 25 PCR cycles. The PCR product was then concentrated using a 30k Amicon filter by spinning at $14,000 \times g$ for 10 min at 4 °C, before subjected to gel purification. After confirming the product bands in denaturing gel as described above, extraction from gel was performed in a manner similar to the 'rapid gel extraction' protocol described by Mayer and Churchman[59]. Briefly, gel pieces containing the product bands were cut out and run through 21 G needle-punctured holes at the bottom of a 0.6-ml microcentrifuge tube by spinning the tube placed inside a 1.5-ml tube at $10,000 \times g$ for 2 min at RT, to crush the gel pieces into thinner bits. Then ~200 µl of TE buffer was added and vortexed to create a gel slurry. After briefly spinning down to settle the mixture, the tube was incubated for 10 min at 70 °C, followed by vortexing and transferring into a Freeze 'N Squeeze tube (Bio-Rad, Cat. No. 7326165), and spinning at $20,000 \times g$ for 3 min at RT to separate PCR products from gel pieces. The product solution was again concentrated using a 30k Amicon filter as described above before sending for sequencing. NGS was performed on an Illumina NovaSeq platform by Theragen Bio, and the sequencing data were analyzed primarily using MATLAB. Only reads containing both primer 'a' and 'b' sequences in full were considered (36,944,830 lines; ~94.08% of total reads), and sequencing errors within the barcode domains were tolerated up to edit distance of 2 from each barcode; reads containing errors beyond the tolerance limits were counted as 'other' (~1.56% in total).

### Fluorescence microscopy
For fluorescence imaging of the cell samples, multi-well chambers containing the cell samples were directly imaged using an inverted epi-fluorescence microscope (Leica Microsystems, DMI6000B), with a 100× oil immersion objective with numerical aperture 1.47 (HCS PL APO 100×/1.47 OIL). After a recording reaction was run as described above, but with the release primer (e.g., 'd*') excluded, each chamber was washed once with 1× ThermoPol buffer and twice with 0.5× TAE/Mg, for 5 min each on a shaker. Then each sample was incubated with a 120 µl solution containing 100 nM imager strands (Atto565-d*-bc*) in 1× TAE/Mg for 30 min on a shaker, followed by washing three times with 0.5× TAE/Mg for 5 min each on a shaker (In the second washing round, DAPI stain was added at 0.2% in the washing buffer to stain the cell nuclei.) Finally, the solution was replaced with 1× TAE/Mg before imaging. Fluorescence images were processed and analyzed using ImageJ.

### Statistics and reproducibility
The statistical significance of the differences in normalized gel intensity between the test groups in the multivalent interaction detection experiments (as shown in Fig. 4d) was examined using two-tailed unequal variance $t$-tests for the measurement data. The resulting $P$ values and statistical parameters are listed in Source Data. The sample size ($n = 8$) was chosen to ensure the differences in gel intensity between groups become statistically significant. For Fig. 4f, g, 35, 32 and 33 different cells were examined, from which 387, 174 and 504 traces were taken, for groups (1), (2) and (3), respectively, across two independent experiments. For other tests, where no comparison between groups is necessary, no statistical method was used to pre-determine repetition size, but all experiments were repeated multiple times, with all attempts yielding consistent results. Specifically, for

data in Fig. 2b, tests were repeated four times; for data in Fig. 2c, tests were repeated at least five times to confirm the results and to use as a reference for other tests; for data in Fig. 2f, AFM imaging for that particular sample was performed once, but similar tests with other patterns (e.g., line; data not shown) were performed multiple times (at least five times) altogether; for data in Fig. 3c, tests were repeated eight times or more with the identical or similar conditions to confirm the results and to use as a reference for other tests; for data in Fig. 3f, tests were repeated five times with the identical or similar conditions; for data in Fig. 3h, tests were repeated at least five times with the identical or similar conditions. No data were excluded from the analyses unless otherwise noted (e.g., NGS reads missing essential primer sequences were excluded as mentioned above). The experiments were not randomized, but where the cell population was divided into three groups for respective treatments, no pre-assignment or selection procedure was applied, and each population was taken from the same source through consecutive and consistent pipetting. Likewise, in the recording experiments, the incubation solutions were prepared from the same master mixes and applied in parallel through consecutive and consistent pipetting. The investigators were not blinded to allocation during experiments and outcome assessment; for most experiments in this study, blinding was not relevant, because the tests were run in parallel (as in gel experiments). For microscopy experiments, blinding was practically not possible, because the sample prep, data acquisition and analysis were performed by the same investigator(s).

## Software

Gel images were obtained on Typhoon FLA 9000 (GE), FluorChem M (ProteinSimple) or Azure C600 (Azure Biosystems) with their respective software provided by the manufacturers (Typhoon software and FluorChem software version numbers unknown; Azure cSeries Capture Software, ver. 1.9.4.0517). AFM images were taken using a Multimode AFM (Veeco Metrology Group) with a Nanoscope V controller and the accompanying software (version number unknown). Quantitative PCR data were acquired using QuantStudio 1 (Thermo Fisher) and processed using the software from the manufacturer (QuantStudio Design & Analysis Software, ver. 1.5.2). Fluorescence microscopy images were obtained from Leica DMI6000B with the software provided (version number unknown). Gel data and fluorescence images were analyzed using ImageJ (ver. 2.1.0) and additional postprocessing (e.g., averaging) was performed using MATLAB (R2021a or R2023b). Averaging of AFM images and plotting of graphs were performed using MATLAB. High-throughput sequencing data were analyzed using a combination of shell commands, MATLAB, and Google Sheets. Statistical tests and curve fitting were performed using MATLAB.

## Reporting summary

Further information on research design is available in the Nature Portfolio Reporting Summary linked to this article.

## Data availability

The high-throughput sequencing data generated in this study have been deposited in the National Center for Biotechnology Information (NCBI) Sequence Read Archive (SRA) database under accession code SRR28124321. Source data are provided with this paper.

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

## Acknowledgements

This work was supported by National Institutes of Health grants (DP1GM133052, R01GM124401) and Wyss Institute for Biologically Inspired Engineering's Molecular Robotics Initiative fund awarded to P.Y., and the National Research Foundation (NRF) Basic Science Research Program grant (RS-2023-00249513) awarded to S.W. by the Ministry of Education of Korea. S.W. acknowledges support from the Damon Runyon Cancer Research Foundation Howard Hughes Medical Institute postdoctoral fellowship (DRG-2203-14) awarded to S.W. and S.K.S. thanks support from the Human Frontier Science Program long-term postdoctoral fellowship (LT000048/2016-L) awarded to S.K.S. The authors thank Allen Zhu and Yu Wang for technical assistance with DNA-conjugated antibodies, Ninning Liu for help with materials transfer, Seung Soo Oh for support with space and resources, and Erik Winfree and Lulu Qian for helpful discussions. Molecular model graphics in Fig. 1 were produced using UCSF Chimera[60] and parts of Fig. 4a, b and e were prepared using Biorender.com.

## Author contributions

S.W. and P.Y. conceived of the study. S.W. and S.K.S. designed and conducted the cell experiments. S.W. designed and performed all other experiments. F.X. participated in the early development of the molecular designs. S.W., S.K.S. and P.Y. analyzed the data. P.Y. supervised the study. S.W. wrote the initial draft and all authors contributed to the final manuscript.

## Competing interests

S.W., F.X. and P.Y. filed patents as inventors through Harvard College to the US Patent and Trademark Office (Pub. No. US11286517B2 & App. No. US20220348990A1), to the European Patent Office (Pub. Nos. EP3916107B1, EP4239079A3 and EP3417078B1), to the World Intellectual Property Organization (Pub. No. WO2017143006A1), to the Japan Patent Office (Pub. No. JP7085999B2), and to the China National Intellectual Property Administration (Pub. Nos. CN108779486B & CN116949140A), which contain the basic mechanism of the crawler operation. S.K.S. is a consulting scientific co-founder of Digital Biology Inc. F.X. is a co-founder of Spear Bio Inc. P.Y. is a co-founder of Ultivue Inc., Spear Bio Inc., and Digital Biology Inc. The authors declare no other competing interests.
