## [Peer Review File · Nature Communications]

REVIEWER COMMENTS

Reviewer #1 (Remarks to the Author):

The paper by Yin and coauthors reports on a set of experiments to demonstrate of simple form of molecular DNA “agent” crawler, which performs on DNA-decorated surfaces and establish a record of its trajectory in its own sequence. The experiments reported are well described and support the claims for the most part. Only relatively trivial trajectories/behaviors are reported, but I the paper still contains sufficiently novel insight in my opinion in this proof-of-concept state. See minor comments for improvements below.

- The use of “swarm” in the title and body is questionable, as in a robotic context “swarm” usually implies a population robots interacting together to perform a task that one could not do alone. This is clearly not the case here, as there is no report of interaction between the robots, nor collective emergent function. Maybe the author are already thinking about something in this direction, which could be discussed in the conclusion.
- Fig 4c: endpoint PCR gel analysis is notoriously not a proper way to evaluate initial target concentration. This should be done by qPCR measurements against a dilution range of control. This is especially important as the authors promote a quantitative approach to biological questions in the conclusion; better lead by the example.
- Fig2e (scalability to lon crawls). Can you also show the PCR gel and sanger sequence? This would better support the scalability claim than a single AFM image or even an averaged version.
- “genetic tagging followed by click coupling”. Can you explain?
- “At the end of a recording round, a ‘record’ molecule gets autonomously released into the solution, and this allows one of the key properties of the crawler system that the recording process is non-destructive to the probes, leaving the state of each probe unchanged after each round, hence allowing catalytic, repeated recording.” and “The use of agents allowed non-destructive and repeated sampling of information from target molecules, enabling amplification of weak signals from low concentration samples while the molecules are kept intact”. I do not think that the “non-destructive” and “repeated sampling” claims are supported here. The design in principle allows this when the release primer is included in the crawling mixture, because every binding site can in principle support more than one crawler extension, but the claims would require a demonstration that more than one turnover is performed (by quantitative analysis of the amount of crawler obtained) and/or a repeated recording process.
- It is unfortunate that only gels are used to analyse the recorded crawling data. The method would clearly become more interesting in combination with deep sequencing readout and more informative design containing UMI or barcodes (as mentioned in the conclusion). Even if I agree that application of the method to real life questions can be left for further study, a toy model such as a extending the

origami demonstration (fig 2e) to a small maze with a alternative paths and decoding the crawler preferences by NGS would have been much welcome.

- “and membrane proteins typically form clusters of certain sizes when initiating signal cascades” this is a bit vague and not well connected to the first part of the sentence (which is itself not very relevant to the discussion).
- “Could provide a route towards recording interactions of numerous individual components inside a cell and thus allowing an extensive map of molecular interactions to be created with nanoscale resolution”. This seems to imply that transient molecular interactions could be recorder by the crawlers, which is not the case here (since recording event also creates a strong DNA link between the two targets)

Reviewer #2 (Remarks to the Author):

The authors presented a smart design of DNA robotic system that surveys molecular environments and reports spatial information. In this design, a proximity-hybridization-driven DNA crawler can get records from the targets, and thereby reflect their trajectories. It's illustrated by the proof-of-concept experiments on the DNA origami and inside cells, where the crawler is employed to count the number of subunits in molecular complexes and distinguish colocalization patterns of cellular proteins. The conclusions are supported by convincing data, and the manuscript was well organized and written. It may be suitable for publication in this journal after addressing some comments:

1. In Fig 4, the authors got trivalent and monovalent records from three proteins inside cells, which shows the assembly or disassembly state of the microtubule. A control system should be supplemented, which can get bivalent records from the heterodimers of alpha and beta tubulins. It's helpful to estimate the microtubule disassembly or out of the growing phase. In addition, can this system record the spatial sequence of three proteins?
2. In Fig 1C, top row, the last two steps should be in an equilibrium, and only undergo entirely with the assistance of another probe nearby (colored in blue in the middle row). It should be reflected in this panel to clearly depict the mechanism of this DNA crawler.
3. What is the distance between two adjacent probes for proximity DNA hybridization? It should be shown in the schemes to understand clearly.

Reviewer #3 (Remarks to the Author):

The manuscript reports a DNA crawler system that surveys spatial information with nanoscale resolution in an autonomous and repeated manner. It is mainly based on sequential DNA extension/strand displacement and sequencing analysis of concatenated DNA records. The authors used this system to distinguish colocalization patterns of three proteins inside fixed cells under different conditions. The work is a significant extension of the authors' previous methods (Nat. Commun. 2017, 8, 696; Nat. Chem. 2017, 10, 155; bioRxiv, DOI: 10.1101/2020.08.27.271072.). It is an interesting study, and the manuscript is well-written. I particularly liked the Introduction and Conclusions section. I recommend the publication with several minor issues:

1. As shown in Fig. 1, this DNA robotic system follows the sequential DNA extension/strand displacement (first a to b, b to c, and then c to d). Both in the abstract and the sixth paragraph, the method demonstrated can 'detect multivalent proximity interactions'. Taking Fig. 1C for example, when the target b is absent, how to simultaneously detect the multivalent proximity interactions involved with a-c, a-d or a-c-d? A potential detection scheme may be provided in the Supplementary Information.
2. In the fifth paragraph, while studying biomolecular interactions based on DNA proximity detection, it is argued that most existing methods are 'destructive'. Therefore, each probe can only be used once, hence potentially leaving 'dead spots' and resulting in incomplete analyses (e.g., in an arrangement A-B-C, when A-B is read, C can remain as a dead spot). However, the proximity nicking method reported by ref. 45 can read all targets (e.g., A, B and C) by each probe. On the other hand, ref. 45 and another paper (Angew. Chem. Int. Ed. 2021, 60, 3428 -3432) reported the recording of chromatin modifications within a nanoenvironment. The differences about molecular environment and proximity recognition principle between these two references and this manuscript should be well discussed in the Introduction or other sections.
3. In Fig. 3, the author used primer binding domain with universal sequences to demonstrate the crawler system is capable of randomly moving around and examining unspecified targets. However, as shown in Fig. 3b, the 'record' molecule always binds to the recorded sites and only released at the end of a recording round. Due to the rigidity of DNA duplex, the restriction of primer became more obvious as the number of recording rounds increase, such as direction and accessibility. So, for the whole crawler system, maybe only the initiation procedure runs in random mode.
4. As shown in Fig. 3c, the yields of reaction products get weaker as the number of counts increase. Maybe the overlong 'record' molecule prone to limit reaction efficiency due to the steric hindrance. So, if the author evaluated the maximum number of recording rounds the system can achieve?
5. In Fig. S6c, we noticed the reaction substrates are far much more than the longest product. Does this result means the low efficiency for the detection of multivalent proximity interactions? As we know, super-resolution fluorescence microscopy presents well performance for in situ detection of nanoscale spatial information despite the limitation of the numbers of fluorescence signals.
6. What's the difference between probe architecture B and E in Fig. S1? Which design is better? Do these designs affect the dynamics of hybridization, strand displacement and releasing?

7. Fig. S2-7 do not exist in the main text.

Response to reviewer comments:

The authors would like to thank the reviewers for their thoughtful and comprehensive comments. They were very helpful in improving the manuscript. We provide our response below, along with some excerpts from the revised manuscript with changes highlighted, where relevant.

Reviewer #1 (Remarks to the Author):

The paper by Yin and coauthors reports on a set of experiments to demonstrate of simple form of molecular DNA “agent” crawler, which performs on DNA-decorated surfaces and establish a record of its trajectory in its own sequence. The experiments reported are well described and support the claims for the most part. Only relatively trivial trajectories/behaviors are reported, but I the paper still contains sufficiently novel insight in my opinion in this proof-of-concept state. See minor comments for improvements below.

- The use of “swarm” in the title and body is questionable, as in a robotic context “swarm” usually implies a population robots interacting together to perform a task that one could not do alone. This is clearly not the case here, as there is no report of interaction between the robots, nor collective emergent function. Maybe the author are already thinking about something in this direction, which could be discussed in the conclusion.

> We appreciate the reviewer’s suggestion. Although we believe that the current manuscript does demonstrate a swarm-like effect, in the sense that while the action of each agent is simple and minimal (crawling around until it is lifted by a release primer) their combined products provide collective answers that each molecule cannot directly offer, such as the number of subunits in molecular complexes and the status of multivalent protein colocalization, we see that the term might still cause confusion. Therefore, we have now added a brief discussion in the conclusion (also copied below), attempting to clarify what we intended and mentioning more active forms of swarming as potential future development, following the reviewer’s suggestion.

In the first paragraph of the Conclusions section (page 6, lines 347-354): “We have demonstrated a system of molecular robotic swarm ‘agents’ for quantitative examination of molecular landscapes. The use of agents allowed non-destructive and repeated sampling of information from target molecules, enabling amplification of weak signals from low concentration samples while the molecules are kept intact. While the task of each agent was simple and minimal, their combined products allowed us to reach ‘collective’ answers such as the number of subunits in molecular complexes and the status of multivalent protein colocalization. With more development, more active forms of ‘swarm’ behaviors could also be designed, by encoding ‘interactions’ between the agents, e.g., through a record generated by one agent becoming an instruction to be read and followed by another agent.”

- Fig 4c: endpoint PCR gel analysis is notoriously not a proper way to evaluate initial target concentration. This should be done by qPCR measurements against a dilution range of control. This is especially important as the authors promote a quantitative approach to biological questions in the conclusion; better lead by the example.

> We agree that a qPCR-based quantification analysis would have been necessary if our purpose had been to count the exact copy numbers of the records generated. However, our main goal in those sets of experiments was to demonstrate that our agents can collectively reveal the colocalization status of multiple proteins. Our emphasis on quantitative understanding in biology discussed in the conclusion was about finding answers such as the size of molecular complexes, which can still be obtained without knowing the exact copy numbers of the individual agents involved in the analysis.

Nonetheless, we do also agree that those extra pieces of data would be helpful to the reader for better understanding of our system. However, unfortunately, performing qPCR analysis for the same samples is not possible at this point and rebuilding the exact same setup is also practically challenging, given that all key authors involved in those sets of experiments have since moved on to different positions in different countries. Instead, to complement this, we ran an additional set of gel-based ‘calibration’ tests, where we prepared record products of the same sequence at a known concentration, created series of dilutions, applied the same PCR protocol, and compared the resulting relative gel band intensities (as normalized by the ladder band intensities in the same gel) to the relative intensities of the bands we provided in the original figure (also normalized by the ladder bands in that gel) to estimate the pre-amplification amounts of the original records. We added a brief discussion in the relevant section of the main text (also copied below), and a figure summarizing the analysis in Supplementary Information (Supplementary Fig. 7).

On page 6, lines 316-319: “..., the level of trivalent records dropped significantly for group (2) and recovered to a level comparable to group (1) after the regrowth treatment for group (3) (28.4% and 93.9%, respectively, of group (1), **comparing gel band intensities**, in the particular test shown in Fig. 4c; **or ~4.7% and ~68.9%, respectively, in pre-amplification record quantity, estimated based on gel-based calibration tests—see Supplementary Fig. 7).**”

- Fig2e (scalability to lon crawls). Can you also show the PCR gel and sanger sequence? This would better support the scalability claim than a single AFM image or even an averaged version.

> To better support scalability of the crawler reaction, we added extra pieces of data in Supplementary Information that were not initially included in the paper, which is a gel image after PCR that shows the records from crawling reactions along a long track, with up to 11 steps (Supplementary Fig. 5). One can see that crawlers reach the full 11 steps, which is primarily limited just by the origami size. We added short descriptions in the main text as well, also copied below.

On page 3, lines 151-152: “Fig. 2f shows such an example where a set of three kinds of probes (each marked ‘b-c’, ‘c-d’, and ‘d-b’) are placed along the boundary of a shape in order and in a repeated

fashion, with a crawler successfully grown along the track. See Supplementary Fig. 5 for an additional set of data that supports scalability of the crawling reaction.”

- “genetic tagging followed by click coupling”. Can you explain?

> Here, we meant to refer to the kinds of methods where click-compatible chemical moieties are genetically encoded for click coupling, typically through unnatural amino acids, as described by e.g., Plass, T. et al. [Genetically encoded copper-free click chemistry. *Angew. Chem. Int. Ed.* **50**, 3878–3881 (2011)], Nikić, I. et al. [Labeling proteins on live mammalian cells using click chemistry. *Nat. Protoc.* **10**, 780–791 (2015)] or Arsić, A. et al. [Minimal genetically encoded tags for fluorescent protein labeling in living neurons. *Nat. Commun.* **13**, 314 (2022)]. But we see that it would be more appropriate to use more general and comprehensive terms, thus changed the phrase to “genetic tagging followed by **chemical coupling**”.

- “At the end of a recording round, a ‘record’ molecule gets autonomously released into the solution, and this allows one of the key properties of the crawler system that the recording process is non-destructive to the probes, leaving the state of each probe unchanged after each round, hence allowing catalytic, repeated recording.” and “The use of agents allowed non-destructive and repeated sampling of information from target molecules, enabling amplification of weak signals from low concentration samples while the molecules are kept intact”. I do not think that the “non-destructive” and “repeated sampling” claims are supported here. The design in principle allows this when the release primer is included in the crawling mixture, because every binding site can in principle support more than one crawler extension, but the claims would require a demonstration that more than one turnover is performed (by quantitative analysis of the amount of crawler obtained) and/or a repeated recording process.

> We appreciate the reviewer’s critical comment. Following the reviewer’s suggestion, we performed additional sets of experiments where the records generated at different recording durations were collected and quantified using qPCR, and then were compared to the initial origami amount added (i.e., the number of probe sets). We included the data in Figure 2d; one can see that the number of records generated per origami exceeds 1-fold within 30 minutes of recording and reaches over ~100-fold after 3 hours, which clearly supports the non-destructive and repeated sampling principle of the design. A brief discussion has been added in the main text as well (copied below).

On page 2, lines 134-138: “A qPCR analysis shows that record generation quickly (within ~30 min) surpassed the number of origami templates added and reached a ~100-fold excess in 3 hr (Fig. 2d) under a certain recording condition, which supports the non-destructive and repeated recording mechanism.”

- It is unfortunate that only gels are used to analyse the recorded crawling data. The method would clearly become more interesting in combination with deep sequencing readout and more informative design

containing UMI or barcodes (as mentioned in the conclusion). Even if I agree that application of the method to real life questions can be left for further study, a toy model such as a extending the origami demonstration (fig 2e) to a small maze with a alternative paths and decoding the crawler preferences by NGS would have been much welcome.

> Following the reviewer's suggestion, we designed and conducted an experiment where crawlers were allowed to take different paths on a simple square track, and analyzed the path preferences using NGS, which has been added in Figure 3. NGS revealed alternate routes that could not be resolved by gel analysis, by unveiling different paths that would have created records of the same length. Interestingly, crawlers did exhibit different 'preferences' for different paths, one prominent trend being that the crawlers preferred a shorter path to a longer one. A paragraph describing the experiment and results has been added in the relevant section (copied below).

On page 3, lines 220-238: "One may note that, in random crawling, a crawler would take a different path each time, as each step involving repeating primers can occur in any direction. Thus, records with the same length produced after the same number of steps can contain information for different trajectories. To test if we can 'decode' the different paths, we designed a simple square-shaped track containing two probes with repeating primers, each labeled with different barcodes (Fig. 3g), and analyzed the records with high-throughput sequencing. Gel data shows that crawlers produce records of three different lengths, as expected from a complex of size 4 with start and finish sites, but both length-3 (len3) and length-4 (len4) records can originate from two different paths, respectively, as depicted in Fig. 3h. Sequencing analysis confirmed that it is indeed the case, and revealed the distribution of different paths taken (Fig. 3i,j). While the ratios between sequencing read counts do not directly represent the actual ratios of record populations especially between different length products, due to potential biases in PCR and sequencing processes, the general trends within len3 and len4 products display the 'preference' of the crawlers for a shorter path over a longer (diagonal) one, likely reflecting kinetic advantage."

- "and membrane proteins typically form clusters of certain sizes when initiating signal cascades" this is a bit vague and not well connected to the first part of the sentence (which is itself not very relevant to the discussion).

> This part was not meant to be conceptually connected to the first part of the sentence. To prevent confusion, we now separated the two parts as individual sentences.

- "Could provide a route towards recording interactions of numerous individual components inside a cell and thus allowing an extensive map of molecular interactions to be created with nanoscale resolution". This seems to imply that transient molecular interactions could be recorder by the crawlers, which is not the case here (since recording event also creates a strong DNA link between the two targets)

> We did not mean to record transient molecular interactions in real time with the current system; however, with improvements in kinetics we believe that in the long run this kind of mechanism could be used to monitor molecular interactions with certain time scales. Nonetheless, we do understand the concern and potential confusion that the statement might cause, so we decided to limit the discussion to ‘fixed’ cells.

On page 6, lines 377-378: “In the long run, with unique labeling and massively-parallel sequencing readouts, the crawler system could provide a route towards recording interactions of numerous individual components, e.g., inside a fixed cell, and thus allowing an extensive map of molecular interactions to be created with nanoscale resolution.”

Reviewer #2 (Remarks to the Author):

The authors presented a smart design of DNA robotic system that surveys molecular environments and reports spatial information. In this design, a proximity-hybridization-driven DNA crawler can get records from the targets, and thereby reflect their trajectories. It’s illustrated by the proof-of-concept experiments on the DNA origami and inside cells, where the crawler is employed to count the number of subunits in molecular complexes and distinguish colocalization patterns of cellular proteins. The conclusions are supported by convincing data, and the manuscript was well organized and written. It may be suitable for publication in this journal after addressing some comments:

1. In Fig 4, the authors got trivalent and monovalent records from three proteins inside cells, which shows the assembly or disassembly state of the microtubule. A control system should be supplemented, which can get bivalent records from the heterodimers of alpha and beta tubulins. It’s helpful to estimate the microtubule disassembly or out of the growing phase. In addition, can this system record the spatial sequence of three proteins?

> We appreciate the reviewer’s suggestion. While the kind of control experiment separately confirming two-protein interactions would have been nice additional data that supports the crawler operation in general, we deem that it may not add much meaning to our particular case, given that alpha and beta tubulins are known to exist in the dimeric form most of the time (see, e.g., Caplow, M. & Fee, L. *Mol. Biol. Cell* **13**, 2120–2131 (2002)) and that the mechanism of nocodazole is known to be to bind the ‘intradimeric’ interface between alpha and beta tubulins to freeze a ‘kinked’ conformation that disfavors assembly *between* the dimers (see, e.g., Chen, G.-Y. et al. *Curr. Biol.* **29**, 2259-2269.e4 (2019)). In other words, it would be generally accepted that nocodazole takes microtubules out of the growing phase by disturbing the ‘interdimeric’ contacts and thus promoting microtubule disassembly, so we are not sure if trying to detect the alpha-beta tubulin dimer would be meaningful in the context of our study. Delving further in and trying to reconfirm the mechanism of nocodazole’s action would be an interesting direction that should be accompanied by other biochemical characterizations, and we believe it is beyond the scope of this paper.

As to recording the spatial sequence of three proteins, we do believe that the crawler system can be used to record the relative arrangements of three or more proteins in proximity, depending on the spacings between the proteins. For example, in the newly added data in Fig. 3, with a square-shaped artificial complex, while the crawler was able to visit all probe sites because they were all within proximity, the NGS-based analysis revealed different preferences depending on the relative distances between probe sites. Applying a similar principle to real protein complexes to decode the relative spatial arrangements would be an interesting topic for future studies.

2. In Fig 1C, top row, the last two steps should be in an equilibrium, and only undergo entirely with the assistance of another probe nearby (colored in blue in the middle row). It should be reflected in this panel to clearly depict the mechanism of this DNA crawler.

> We appreciate the reviewer's critical eye. We agree that the mentioned steps should be reversible, and likewise the first step of the reaction where a primer binds the primer-binding site must be reversible, too, while it would favor the bound state at room temperature. We thus replaced the single-sided arrows in the relevant locations with double arrows to clarify equilibrium reactions.

3. What is the distance between two adjacent probes for proximity DNA hybridization? It should be shown in the schemes to understand clearly.

> The actual distances between probe sites used in different tests in the paper can be found in the probe position maps included in Supplementary Information (see Supplementary Fig. 2 in the revised version), and the distance is tunable by changing the length of the 'spacer' domain of the probes, as demonstrated in a similar system in our previous work (see Fig. 4c in Schaus, T. E. et al, A DNA nanoscope via auto-cycling proximity recording. *Nat. Commun.* **8**, 696 (2017)). We do, however, agree that giving a rough idea would be helpful to the reader, and thus added a brief description in the caption for Fig. 1 (also copied below).

In the caption for Figure 1: "The new primer can initiate a next reaction, as shown in the middle row, with another probe nearby (typically within tens of nanometers; tunable)."

Reviewer #3 (Remarks to the Author):

The manuscript reports a DNA crawler system that surveys spatial information with nanoscale resolution in an autonomous and repeated manner. It is mainly based on sequential DNA extension/strand displacement and sequencing analysis of concatenated DNA records. The authors used this system to distinguish colocalization patterns of three proteins inside fixed cells under different conditions. The work

is a significant extension of the authors' previous methods (Nat. Commun. 2017, 8, 696; Nat. Chem. 2017, 10, 155; bioRxiv, DOI: 10.1101/2020.08.27.271072.). It is an interesting study, and the manuscript is well-written. I particularly liked the Introduction and Conclusions section. I recommend the publication with several minor issues:

1. As shown in Fig. 1, this DNA robotic system follows the sequential DNA extension/strand displacement (first a to b, b to c, and then c to d). Both in the abstract and the sixth paragraph, the method demonstrated can 'detect multivalent proximity interactions'. Taking Fig. 1C for example, when the target b is absent, how to simultaneously detect the multivalent proximity interactions involved with a-c, a-d or a-c-d? A potential detection scheme may be provided in the Supplementary Information.

> We appreciate the reviewer's critical comment. The reviewer is correct that the model case shown in Fig. 1 describes a 'sequential' recording scheme where the crawler must follow prescribed paths, and thus the absence of the middle connecting primer (e.g., 'b') will prevent the reaction from completing. But we would like to emphasize that this scheme of sequential recording can still distinguish the colocalization of multiple proteins (e.g., A, B, and C, with a-b, b-c, and c-d primer sets, respectively) if they are all in close proximity (or if they are sequentially positioned) from a state where they are separated, as demonstrated for the microtubule growing ends as shown in Fig. 4. We do, however, admit that it is certainly limited in that the scheme cannot distinguish just A and C colocalizing without B versus all being separated.

An alternative way to allow detection of any spatial organizations beyond sequential is by using probes with 'repeating primers' (as described in Figure 3 and the related section) such that a crawler can crawl to any site without a specified order, and analyzing the records with high-throughput sequencing. This direction is already suggested in Conclusions.

2. In the fifth paragraph, while studying biomolecular interactions based on DNA proximity detection, it is argued that most existing methods are 'destructive'. Therefore, each probe can only be used once, hence potentially leaving 'dead spots' and resulting in incomplete analyses (e.g., in an arrangement A-B-C, when A-B is read, C can remain as a dead spot). However, the proximity nicking method reported by ref. 45 can read all targets (e.g., A, B and C) by each probe. On the other hand, ref. 45 and another paper (Angew. Chem. Int. Ed. 2021, 60, 3428 -3432) reported the recording of chromatin modifications within a nanoenvironment. The differences about molecular environment and proximity recognition principle between these two references and this manuscript should be well discussed in the Introduction or other sections.

> We thank the reviewer for pointing us to the work reported in *Angew. Chem.*; we had missed this interesting work. While the work, now cited as ref. 46, and ref. 45 both do cleverly circumvent the problem of 'dead spots', by 'rescuing' the spots left unligated after proximity ligation (ref. 46) or by activating all nearby probes from a central walker (ref. 45), we still believe these methods can be classified as *destructive* methods in that probes once used for one measurement cannot be reused for another when it becomes necessary. For example, for ref. 46, after detecting A-B proximity (and saving those 'lone' A and B sites), if another pair of partners need to be detected (e.g., A-C), then we suppose

those A sites cannot be reused, unless there is a systematic intervention step from a researcher such as thermal cycling. For ref. 45, while all A, B, and C colocalization sites can be activated from a central probe (call it 'x') in one round, if a subsequent round is necessary from another walker probe (say, 'y') that is surrounded by, e.g., A, D and E, then the A probe cannot produce meaningful data because it has already gone through an irreversible change from the previous round and will remain activated, now becoming a permanently bright 'white pixel'.

We agree that discussion on these subtle differences is important and would be interesting to the reader, thus we have included it in a concise form in the main text, where we first introduce those works in Introduction.

On page 1, lines 62-67: "..., hence potentially leaving 'dead spots' and resulting in incomplete analyses (e.g., in an arrangement A-B-C, when A-B is read, C can remain as a dead spot; the proximity nicking⁴⁵ method and another recent ligation-based method⁴⁶ cleverly circumvent this limitation by activating all nearby spots surrounding a central probe or by 'rescuing' unligated spots, respectively, but they are still destructive methods in that probes once used cannot be reused for subsequent assays with different partners when needed)."

3. In Fig. 3, the author used primer binding domain with universal sequences to demonstrate the crawler system is capable of randomly moving around and examining unspecified targets. However, as shown in Fig. 3b, the 'record' molecule always binds to the recorded sites and only released at the end of a recording round. Due to the rigidity of DNA duplex, the restriction of primer became more obvious as the number of recording rounds increase, such as direction and accessibility. So, for the whole crawler system, maybe only the initiation procedure runs in random mode.

> We believe this comment touches upon two separate aspects. One is whether the crawler can keep progressing to nearby probe sites when the motion might become limited due to structural constraints as the number of sites to visit increases, and the other is whether real random investigation is possible with the current design when apparently only the initiation step can be considered completely random.

As to the first point, while it may be true that the motion of the crawler can be partially restricted by previous steps, we would like to emphasize that the crawler *is* capable of crawling over multiple (> ~10) probe sites, because each step is, in principle, just a local reaction between the two terminal probe sites, as shown in our 'scalability' data in Fig. 2f (used to be 2e before revision) as well as in the newly added data in Supplementary Fig. 5.

Regarding the second concern, while it is true that after the initiation step the next move of the crawler is limited to just nearby sites, we can stress that which site to proceed to is also completely random if we do not consider complicated factors such as structural heterogeneity (which one is closer, etc.), and that the reaction can proceed until the crawler is lifted by a release primer at any site (there is no designated "end of a recording round" in random crawling). In addition, even *if* the first point above is a valid issue (random navigation around the initiation spot not proceeding much due to structural constraints), since the

initiation step can happen at any site, the collective actions of multiple crawlers will be able to cover and examine different sites of unspecified targets over a wide area.

4. As shown in Fig. 3c, the yields of reaction products get weaker as the number of counts increase. Maybe the overlong ‘record’ molecule prone to limit reaction efficiency due to the steric hindrance. So, if the author evaluated the maximum number of recording rounds the system can achieve?

> We believe our response to point 3 above partially addresses this concern as well. As shown in Supplementary Fig. 5, we do not think there is an inherent limit in the number of steps that can be achieved as long as the resources for the reaction are supplied, and the current maximum we show (~11 steps) is primarily due to the limit in nanoscale space in the template structure used.

5. In Fig. S6c, we noticed the reaction substrates are far much more than the longest product. Does this result means the low efficiency for the detection of multivalent proximity interactions? As we know, super-resolution fluorescence microscopy presents well performance for in situ detection of nanoscale spatial information despite the limitation of the numbers of fluorescence signals.

> In this particular example, the production of longer records indeed seemed less efficient. However, with design optimization, kinetics of the crawling reaction seems to improve, which is in part supported by the newly added data in Supplementary Fig. 5, where the crawler was able to crawl multiple steps efficiently. For optimization, some design elements are currently under investigation, and we believe it can be a topic for a future report.

Super-resolution fluorescence microscopy certainly has its own merits, but in addition to the ‘spectral crowding’ issue that the reviewer also pointed out, imaging-based methods share the general limitations that they require samples to be immobilized on a substrate and that they typically suffer limited throughput as only a limited population within a field of view is examined at a time, whereas in our method the agents can examine target molecules free in solution, and do so for the whole population of them at the same time. We did not include this kind of discussion in the current manuscript, because with the current demonstration we cannot yet argue that it directly outperforms microscopy in the aspects mentioned above, but it can be a topic of a future study, especially with high-throughput demonstrations with large-scale sequencing.

6. What’s the difference between probe architecture B and E in Fig. S1? Which design is better? Do these designs affect the dynamics of hybridization, strand displacement and releasing?

> They are different design variations introduced for improved modularity over the course of the development of the system, as described in the figure caption. We do not think that we have tested enough

to compare their effects on dynamics and efficiency, and we believe it is a good subject for a follow-up study.

7. Fig. S2-7 do not exist in the main text.

> We appreciate the reviewer for pointing this out. We have now added descriptions of all the Supplementary items in the main text.

On page 2, lines 120-134: “To demonstrate the basic mechanism of the molecular crawlers, we built a three-point track in triangular alignment on a DNA origami¹⁹ platform (see **Materials and Methods in Supplementary Information for experimental procedures, Supplementary Figs. 1 and 2 for the sequence diagram and probe position maps, respectively, and Supplementary Tables 1-3 for sequence lists**). Fig. 2a depicts a schematic of the design (top) and a rough diagram of a crawler after crawling over the three probes (bottom). DNA origami were deposited on a mica substrate to prevent damage by the polymerase⁴⁷ and to allow atomic force microscope (AFM) characterizations. Fig. 2b shows models and averaged (across different molecules) AFM images of the origami before and ~1 hr after crawlers were allowed to trail the three probes (see **Supplementary Fig. 3 for AFM images prior to averaging**). The fully grown crawler record reaches a length of 100 nucleotides. When the records were retrieved, amplified by PCR and run on a denaturing gel, the final records appear at the expected length range (Fig. 2c; see **Supplementary Fig. 4 for full-height gel images**).”

REVIEWERS' COMMENTS

Reviewer #1 (Remarks to the Author):

The authors have satisfyingly addressed the technical comments and I think the additional Fig2d and 3g-j nicely contribute to the paper.

However, I am not convinced by the answer to the "swarm" comment. It is not because the product of many agents is here "combined" by the authors to produce a readable output that we can say the agents acted "collectively" or as a "swarm" to do something they could not have done alone. Here, if (in a thought experiment) the DNA strands would be sent one by one to the target system, and collected back before the next one is sent, the results, both in counting and in colocalization, would be exactly the same. In fact "collective behaviour" and "swarm robotics" have quite precise interpretation in the respective scientific communities. Copied from wikipedia: "Swarm robotics is an approach to the *coordination* of multiple robots as a system which consist of large numbers of mostly simple physical robots. "In a robot swarm, the collective behavior of the robots results from local *interactions between the robots* and between the robots and the environment in which they act." [1] It is supposed that a desired *collective behavior* emerges from the *interactions* between the robots...").

These interactions are clearly not present in the submitted paper, so I would like to maintain my recommendation not to claim a "Molecular robotic swarm" demonstration and to preserve this terminology for future scientific work actually showing such functionality. Also the addition of "collective" in the new version is unfortunate and will only add to the confusion ("their combined products allowed us to reach 'collective' answers").

Assuming this issue is fixed, I recommend publication.

Reviewer #2 (Remarks to the Author):

The authors have fully addressed the reviewers' concerns. The present version is suitable for publication.

Reviewer #3 (Remarks to the Author):

I am satisfied with the revision and suggest its publication in Nature Commnications.

Response to reviewer comments:

Reviewer #1 (Remarks to the Author):

The authors have satisfyingly addressed the technical comments and I think the additional Fig2d and 3g-j nicely contribute to the paper.

However, I am not convinced by the answer to the “swarm” comment. It is not because the product of many agents is here “combined” by the authors to produce a readable output that we can say the agents acted “collectively” or as a “swarm” to do something they could not have done alone. Here, if (in a thought experiment) the DNA strands would be sent one by one to the target system, and collected back before the next one is sent, the results, both in counting and in colocalization, would be exactly the same. In fact “collective behaviour” and “swarm robotics” have quite precise interpretation in the respective scientific communities. Copied from wikipedia: “Swarm robotics is an approach to the *coordination* of multiple robots as a system which consist of large numbers of mostly simple physical robots. “In a robot swarm, the collective behavior of the robots results from local *interactions between the robots* and between the robots and the environment in which they act.”[1] It is supposed that a desired *collective behavior* emerges from the *interactions* between the robots...”).

These interactions are clearly not present in the submitted paper, so I would like to maintain my recommendation not to claim a “Molecular robotic swarm” demonstration and to preserve this terminology for future scientific work actually showing such functionality. Also the addition of “collective” in the new version is unfortunate and will only add to the confusion (“their combined products allowed us to reach ‘collective’ answers”).

Assuming this issue is fixed, I recommend publication.

> We appreciate the reviewer’s critical comment. We do understand the concern that the reviewer raised, and thus we have now removed the word ‘swarm’ and related discussion throughout our manuscript, except in the Discussion section (renamed from Conclusions, in compliance to the editorial policy), where we provide a future perspective. We copy the relevant part below:

In the first paragraph of the Discussion section: “We have demonstrated a system of molecular robotic ‘agents’ for quantitative examination of molecular landscapes. The use of agents allowed non-destructive and repeated sampling of information from target molecules, enabling amplification of weak signals from low concentration samples while the molecules are kept intact. While the task of each agent was simple and minimal, their combined products allowed us to reach answers that one agent cannot directly provide, such as the number of subunits in molecular complexes and the status of multivalent protein colocalization. With more development, some form of ‘swarm’ behaviors could also be designed, by encoding ‘interactions’ between the agents, e.g., through a record generated by one agent becoming an instruction to be read and followed by another agent.”

Reviewer #2 (Remarks to the Author):

The authors have fully addressed the reviewers' concerns. The present version is suitable for publication.

> We are glad that the reviewer now feels that the manuscript is suitable for publication. We thank the reviewer for the helpful comments again.

Reviewer #3 (Remarks to the Author):

I am satisfied with the revision and suggest its publication in Nature Communications.

> We are glad that the reviewer is now satisfied with the manuscript. We thank the reviewer for the helpful comments again.